# A Study on a Radio Source Location Estimation System Using High Altitude Platform Stations (HAPS)

**DOI:** 10.3390/s24175803

**Published:** 2024-09-06

**Authors:** Yuta Furuse, Gia Khanh Tran

**Affiliations:** Department of Electronic Engineering, Tokyo Institute of Technology, 2-12-1 Ookayama, Meguro-ku, Tokyo 152-8550, Japan; furuse.y.ab@m.titech.ac.jp

**Keywords:** radio emitter, localization, HAPS, sensors, estimation accuracy

## Abstract

Currently, there is a system in Japan to detect illegal radio transmitting sources, known as the DEURAS system. Even though crackdowns on illegal radio stations are conducted on a regular basis every year, the number of illegal emission cases still tends to increase, as ordinary citizens are now able to handle advanced wireless communication technologies, e.g., via software-defined radio. However, the current surveillance system may not be able to accurately detect the source in areas where large buildings are densely packed, such as urban areas, due to the effects of reflected waves. Therefore, in this study, we proposed a system for estimating the location of the source of transmission using a high-flying unmanned aerial vehicle called HAPS. The simulation results using numerical analysis software show that the proposed system can estimate the location of the source over a wider area and with higher accuracy than conventional monitoring systems.

## 1. Introduction

In recent years, mobile communication technology has made great progress, and is used not only in public places but also in our everyday life. For example, mobile communication technology is indispensable in the field of IoT, which is used in smart phones and electrical appliances that everyone has today, and in the field of automatic driving. In fact, the penetration rate of mobile devices is increasing every year [1]. The number of wireless stations has been increasing along with this growth, and as of the end of September 2022, a total of nearly 3 million stations were in operation nationwide in Japan, as shown in Figure 1 [2].

As time has progressed, technologies such as cell phones and IoT have been developed, and mobile communication devices have become more familiar to us, the general public. However, perhaps due to these developments, the number of illegal radio stations (which use power in excess of the limit or illegal frequencies without a license) has been increasing in recent years, as shown in Figure 2 [3].

Today, these illegal radio stations are searched in various ways, and one of them is called a radio wave monitoring system. This system monitors radio waves received at sensor stations and estimates the direction of the radio wave source. Based on this, the location of illegal radio stations is identified by remotely operating sensor stations installed at various locations and sensor stations mounted on vehicles, from a center station installed at each general communications station [3]. However, in urban areas where many buildings are densely built or where there is a lot of vegetation, these systems may not work well because there may be no line of sight from the sensor station to the illegal radio station, or many reflected waves may be generated. In recent years, research has been conducted using unmanned aerial vehicles (UAVs) such as drones to gain some altitude in order to improve the location estimation accuracy of the source of radio waves by making it easier to see the source of radio transmission [4]. However, the service area of this system is limited by the height of the UAVs and actual implementation has not yet been achieved. As for radio monitoring technology, advanced spectrum sensing techniques are still being developed to enhance the detection of illegal radio transmissions in harsh environments [5]. Many other studies on radio monitoring technology [6,7] still exist today.

To address this issue, this paper focuses on the use of the High Altitude Platform System (HAPS) which is a generic term for a system that provides communication services over a wide area by operating an unmanned aircraft, such as an aircraft flown in the stratosphere, like a communication base station [8,9]. As explained here, the HAPS mainly operates unmanned aircraft as base stations, and papers such as “Research and Development of HAPS Mobile Communication System for Rapid Disaster Recovery” [10] have been published. The HAPS has the advantage of being able to deploy unmanned base stations regardless of ground traffic conditions or environments such as the sea. So far, research has been conducted with the idea of operating unmanned aircraft as the main base station. However, if the HAPS is operated as an illegal radio surveillance system, it will be possible to monitor a wider area with less influence from reflected waves and at a lower cost than conventional surveillance systems. If the HAPS can also be operated as a base station in times of disaster, it will be effective in estimating the location of mobile terminals in times of disaster.

This paper examines the operation of the HAPS as an illegal radio monitoring system and a location estimation system for mobile terminals during disasters, and demonstrates its effectiveness mainly through numerical simulations using MATLAB [11]. This paper aims to evaluate the performance of a location estimation system using the HAPS for illegal wireless stations and mobile terminals during disasters that occur in various environments and distances through numerical simulations.

## 2. Methods for Estimating the Location of Radio Transmission Sources

There are two types of systems for estimating the location of radio wave sources such as illegal radio waves and mobile terminals: geometric methods that use various numerical data observed from actual radio wave sources to estimate the location, and statistical methods that estimate the location by comparing the actual observation data with a database obtained using a virtual transmitter in advance. This section describes the geometrical methods currently used to estimate the location of a radio transmitter. In Japan, the DEURAS system [3] is currently used to monitor illegal radio stations. The DEURAS system uses phase differences of the received signals at different antennas to estimate the direction of incoming radio waves. This section briefly explains the methods based on estimation of angle of arrival. 

### 2.1. Estimation Method Using Phase Difference of Received Signals

This section revisits a well-known method for estimating the location of a radio wave source. The direction of arrival of a radio wave is determined by using the phase difference between received signals at multiple elements based on the same principle as that of interferometry. For simplicity, we will explain the estimation of the direction of arrival when a plane wave is incident on two array antennas, as shown in Figure 3. When considering such a situation, the phase difference ∆ϕ between the received signals of arrays 1 and 2 can be expressed, as in Equation (1) [12].
(1)$$\Delta \phi =\frac{2\pi }{\lambda }d_{array}\sin\theta$$

If the complex received signal series of each antenna are $x_{1}$ and $x_{2}$, respectively, the phase difference ∆ϕ can be expressed as in Equation (2) [12].
(2)$$∆\phi =\tan^{-1}\frac{Im\{E[x_{1}x_{2}^{*}]\}}{Re\{E{[x}_{1}x_{2}^{*}]\}}$$
where E[·] denotes an ensemble average operation, λ is the wavelength of the carrier wave, $d_{array}$ is the antenna spacing, Re{·} donotes the real part, and Im{·} denotes imaginary part. When the antenna spacing is 1/2 of the carrier wave wavelength, $\theta =\sin^{-1}\left(\frac{∆\phi }{\pi }\right)$ from Equation (1), and the direction of arrival θ can be obtained in the range $-\frac{\pi }{2}<\theta <\frac{\pi }{2}$ by determining the phase difference ∆ϕ. In a previous study [13], in order to improve the estimation accuracy, the direction of arrival was obtained by using several pairs of two-element arrays to obtain (in advance) the complex reception response $a_{i}(\theta )$ to the arriving wave from the angle θ of element i and introducing the evaluation function shown in Equation (3) [12]. When P(θ) reaches its maximum, θ is consistent with the direction of arrival.
(3)$$P\left(\theta \right)=\frac{1}{{\left|\frac{E\left[x_{1}x_{2}^{*}\right]}{\sqrt{E\left[|x_{1}{|}^{2}\right]E\left[|x_{2}{|}^{2}\right]}}-\frac{a_{1}\left(\theta \right)a_{2}{\left(\theta \right)}^{*}}{\left|a_{1}\left(\theta \right)\right|\left|a_{2}\left(\theta \right)\right|}\right|}^{2}}$$
(4)$$[a_{1}\left(\theta \right),a_{2}\left(\theta \right)]=\left[1,exp\left(-\frac{2\pi d_{array}}{\lambda }\sin\theta \right)\right]$$

The estimated direction of arrival is then used to estimate the location of the radio source by triangulation, as shown in Figure 4.

This method provides highly accurate estimation when there is a line of sight (LoS) from the antenna to the source of the radio wave. However, when estimating the direction of arrival in urban areas, multiple waves are received due to reflected waves. The problem with this method is that the direction of arrival cannot be estimated properly at that time.

### 2.2. Estimation Method Using the Beamformer Method

The Beamformer method is a basic method for estimating the direction of arrival. It estimates the direction of a radio source by scanning the main lobe of a directional antenna with high gain in all directions, as shown in Figure 5, outputting the output voltage change of the array at each angle, and obtaining the maximum output value.

For simplicity, the estimation of the direction of arrival using the Beamformer method is described in detail using a K-element linear array antenna, as shown in Figure 6 [12].

In this case, the output of the array is y(t) can be expressed as follows [12].
(5)$$y\left(t\right)=W^{H}X\left(t\right)$$
(6)$$X\left(t\right)={\left[x_{1}\left(t\right),x_{2}\left(t\right),\ldots ,x_{K}\left(t\right)\right]}^{T}$$
(7)$$W={\left[w_{1},w_{2},\ldots ,w_{K}\right]}^{T}$$
${\left(\cdot \right)}^{H}$ denotes the transpose operation, where the rows and columns of a matrix are swapped. ${\left(\cdot \right)}^{T}$ denotes the Hermitian transpose operation, where the matrix is first transposed and then each element is replaced by its complex conjugate. The output voltage $P_{out}$ is provided as follows [12].
(8)$$P_{out}=\frac{1}{2}E\left[{\left|y\left(t\right)\right|}^{2}\right]=\frac{1}{2}W^{H}R_{xx}W$$
(9)$$R_{xx}=E\left[X\left(t\right)X^{H}\left(t\right)\right]$$

Assuming that L waves are incident on the incoming wave and that the respective signal waveforms and angles of arrival are $F_{l}\left(t\right)$ and $\theta_{l}\left(l=1,2,\ldots ,L\right)$, the respective direction vectors $V_{l}$ can be expressed as follows [12].
(10)$$V_{l}={\left[exp\left(-j\frac{2\pi }{\lambda }d_{1}\sin\theta_{l}\right),\ldots ,exp\left(-j\frac{2\pi }{\lambda }d_{K}\sin\theta_{l}\right)\right]}^{T}\equiv a\left(\theta_{l}\right)$$

Also, $d_{k}(k=1,2,\ldots ,K)$ is the distance from the reference array to the position of the *k*-th element. At this time, the input vector is expressed as follows [12].
(11)$$X\left(t\right)=\sum_{l=1}^{L}F_{l}\left(t\right)a\left(\theta_{l}\right)+N\left(t\right)$$
(12)$$=AF\left(t\right)+N\left(t\right)$$
(13)$$A=\left[a\left(\theta_{1}\right),a\left(\theta_{2}\right),\ldots ,a\left(\theta_{L}\right)\right]$$
(14)$$F\left(t\right)={\left[F_{1}\left(t\right),F_{2}\left(t\right),\ldots ,F_{L}\left(t\right)\right]}^{T}$$

In the above equation, $N\left(t\right)$ is the thermal noise vector, whose components are independent complex Gaussian processes with zero mean and variance (power) $\sigma^{2}\left(=P_{n}\right)$. Also, the correlation matrix $R_{xx}$ is expressed as follows [12].
$$R_{xx}=E\left[X\left(t\right)X^{H}\left(t\right)\right]$$
$$=AE\left[F\left(t\right)F^{H}\left(t\right)\right]A^{H}+E\left[N\left(t\right)N^{H}\left(t\right)\right]$$
(15)$$=ASA^{H}+\sigma^{2}I$$
(16)$$S≜E\left[F\left(t\right)F^{H}\left(t\right)\right]$$

The matrix S in Equation (16), which represents the correlation between arriving waves, is called the signal correlation matrix and is a diagonal matrix with $P_{l}\left(l=1,2,\ldots ,L\right)$ (each input power) in the diagonal components, if all arriving waves are uncorrelated with each other. In order to direct the main lobe of the array antenna to angle θ using the above equation and others, the weight $w_{k}$ expressed in Equation (17) [12] can be multiplied to each input signal based on the common-phase condition (the condition to align the phases to be in-phase).
(17)$$w_{k}=exp\left(-j\frac{2\pi }{\lambda }d_{k}\sin\theta \right)\;\;\;\;\;\left(k=1,2,\ldots ,K\right)$$

The angle θ is varied from −π/2 to π/2 to find the peak of the output voltage of the array. The output voltage of the array is expressed by Equation (18), and the angular distribution by the Beamformer method is obtained by normalizing Equation (18) [12].
(18)$$P_{BH}\left(\theta \right)=\frac{P_{out}}{a^{H}\left(\theta \right)a\left(\theta \right)/2}=\frac{a^{H}\left(\theta \right)R_{xx}a\left(\theta \right)}{a^{H}\left(\theta \right)a\left(\theta \right)}$$

The Beamformer method estimates the direction of arrival and the input power of the arriving wave from the position of the peak of the evaluation function, as shown in Equation (18) using the input correlation matrix $R_{xx}$ and the weight expressed in Equation (17). Using the estimated direction of arrival, the radio wave source is estimated by triangulation, as shown in Figure 4. However, in the case of multiple incoming radio waves, if the main lobe of the antenna pattern shown in Figure 5 faces a certain radio wave while the side lobes (in the direction of relatively high gain) coincide with the direction of other radio waves, the output voltage will include not only one wave but also other waves, and it is not guaranteed that the evaluation function may always have a peak in the desired direction of arrival. In addition, when radio waves with almost similar arrival angles are incident, it may not be possible to distinguish the two waves when scanning with the main lobe having a wide angle of arrival, as shown in Figure 5. In such cases, the angular resolution of location estimation by the Beamformer method becomes low, and the peak value may not accurately represent the power of the signal.

In fact, as shown in [14] which studied the direction of arrival estimation using the Beamformer method in a multipath environment, the estimation accuracy is degraded depending on the number of arriving waves. There is also the Capon method, which directs the main lobe in one direction while at the same time attempting to reduce the output contribution from other directions. However, all of these methods use the main lobe of the antenna pattern, and the angular resolution of the estimation is limited. Therefore, it is difficult to apply the Beamformer method to the scenarios of ground emitters in urban areas where many radio waves arrive.

### 2.3. Estimation Method Using the MUSIC Method

This section describes in detail the MUSIC method, which uses eigenvalues and eigenvectors to estimate the direction of arrival at a higher angle resolution than the Beamformer method [12]. For the sake of simplicity, let assume a K-element linear array antenna is used, as shown in Figure 6. In this case, the input vector X(t) of each antenna is the same as in Equation (12).
(19)$$X\left(t\right)=AF\left(t\right)+N\left(t\right)$$

The correlation matrix of the input vectors is then as in Equation (15).
(20)$$R_{xx}≜E\left[X\left(t\right)X^{H}\left(t\right)\right]=ASA^{H}+\sigma^{2}I$$
where the signal correlation matrix S can be expressed as follows.
(21)$$S=\left[\begin{array}{cccc} E\left[{\left|F_{1}\left(t\right)\right|}^{2}\right] & E\left[F_{1}\left(t\right)F_{2}^{*}\left(t\right)\right] & \cdots & E\left[F_{1}\left(t\right)F_{L}^{*}\left(t\right)\right]\\ E\left[F_{2}\left(t\right)F_{1}^{*}\left(t\right)\right] & E\left[{\left|F_{2}\left(t\right)\right|}^{2}\right] & \cdots & E\left[F_{2}\left(t\right)F_{L}^{*}\left(t\right)\right]\\ \vdots & \vdots & \ddots & \vdots \\ E\left[F_{L}\left(t\right)F_{1}^{*}\left(t\right)\right] & E\left[F_{L}\left(t\right)F_{2}^{*}\left(t\right)\right] & \cdots & E\left[{\left|F_{L}\left(t\right)\right|}^{2}\right] \end{array}\right]$$

In the absence of thermal noise, if the arriving waves are uncorrelated, S becomes a diagonal matrix and its rank is L. If the direction matrix A also differs from the direction of arrival, its column vectors become independent, and its rank becomes L. Therefore, in this case, the correlation matrix $R_{xx}$ is a non-negative definite Hermitian matrix of rank L. Denoting the eigenvalues of this matrix by $\mu_{i}\left(i=1,2,\ldots ,K\right)$ and the corresponding eigenvectors by $e_{i}\left(1,2,\ldots ,K\right)$ [12],
(22)$$ASA^{H}e_{i}=\mu_{i}e_{i}\left(i=1,2,\ldots K\right)$$
and each eigenvalue can be expressed as a real number [12].
(23)$$\mu_{1}\geq \mu_{2}\geq \cdots \mu_{L}>\mu_{L}+1=\cdots =\mu_{K}=0$$

The eigenvectors are orthogonal to each other, as follows [12].
(24)$$e_{i}^{H}e_{k}=\delta_{ik}\;\;\;\left(i,k=1,2,\ldots ,K\right)$$

In the presence of thermal noise, the thermal noise power is added to the eigenvalues and the eigenvectors remain unchanged [12].
$$R_{xx}e_{i}=\left(ASA^{H}+\sigma^{2}I\right)e_{i}$$
$$=ASA^{H}e_{i}+\sigma^{2}e_{i}$$
$$=\mu_{i}e_{i}+\sigma^{2}e_{i}$$
(25)$$=\left(\mu_{i}+\sigma^{2}\right)e_{i}\;\;\;\;(i=1,2,\ldots K)$$

Then we have:(26)$$\lambda_{i}≜\mu_{i}+\sigma^{2}\;\;\;\;\;\;\;\;\left(i=1,2,\ldots ,K\right)$$

Denoting the eigenvalues of the correlation matrix by:(27)$$\lambda_{1}\geq \lambda_{2}\geq \cdots \geq \lambda_{L}>\lambda_{L+1}=\cdots =\lambda_{K}=\sigma^{2}$$
the number of arriving waves L can be obtained by finding the number of eigenvalues of the correlation matrix that have values larger than the noise power. Also, if we focus on the eigenvectors corresponding to the eigenvalues equal to the thermal noise power:(28)$$R_{xx}e_{i}=\left(ASA^{H}+\sigma^{2}I\right)e_{i}=\lambda_{i}e_{i}=\sigma^{2}e_{i}\;\;\;\;\;\;\left(i=L+1,\ldots ,K\right)$$
based on Equation (20), we can derive the following property [12].
(29)$$ASA^{H}e_{i}=0\;\;\;\;\;\;\;\left(i=L+1,\ldots ,K\right)$$

Also, from the fact that both A and S are full ranks.
(30)$$A^{H}e_{i}=0\;\;\;\;\;\left(i=L+1,\ldots ,K\right)$$

In other words:(31)$$a^{H}\left(\theta_{l}\right)e_{i}=0\;\;\;\;\;\;\left(l=1,2,\ldots ,L;i=L+1,\ldots ,K\right)$$

From this equation, it can be seen that the eigenvectors corresponding to eigenvalues equal to the thermal noise power are orthogonal to the direction vectors of all arriving waves. Using this property, the evaluation function of the MUSIC method can be expressed as follows [12] with normalization.
(32)$$P_{M}U\left(\theta \right)≜\frac{1}{\sum_{i=L+1}^{K}{\left|e_{i}^{H}a\left(\theta \right)\right|}^{2}}\times a^{H}\left(\theta \right)a\left(\theta \right)$$
(33)$$=\frac{a^{H}\left(\theta \right)a\left(\theta \right)}{a^{H}\left(\theta \right)E_{N}E_{N}^{H}a\left(\theta \right)}$$
(34)$$E_{N}≜\left[e_{L+1},\ldots ,e_{K}\right]$$

The denominator part of Equation (32) shows a minimum value at θ, which is the direction vector (direction vector of the arriving wave) orthogonal to the eigenvector corresponding to the noise, as mentioned earlier. The MUSIC method, on the other hand, uses the noise portion (the portion where the power is small) to achieve high resolution estimation. However, when multiple radio waves are incident from a single source due to reflected waves, etc., there is a correlation between the incoming waves. In this case, the rank of the signal correlation matrix expressed in Equation (21) is not equal to the number of arriving waves, and the direction of arrival cannot be accurately estimated.

It is known that all of the three location estimation methods using direction of arrival estimation introduced in this section have the problem that the estimation accuracy is degraded by many reflected waves when a ground antenna is used.

## 3. Numerical Analysis and Results

Currently, location estimation using the HAPS is not yet available and is considered to be operated as a base station. Because of this situation, no demonstration experiment of location estimation using the HAPS has been realized. In this section, two methods of direction of arrival estimation will be applied for the HAPS for the location estimation of ground emitters. Since the phase difference method is just a primitive version of the Beamformer one, only the Beamformer and MUSIC methods are introduced in this section for evaluation of the localization accuracy.

### 3.1. Comparison of LoS Rate between the HAPS and UAVs

In Section 2, we discussed various location estimation methods. However, the accuracy of such methods varies greatly depending on whether the environment between the receiving antenna and the radio source is a LoS environment or an NLoS environment. In this section, we perform a ray tracing simulation using the SBR (Shooting and Bouncing Rays method) in MATLAB to compare the LoS rate between the case using the HAPS and the case using a UAV. In this analysis, the case in which a direct wave (a radio wave that is not reflected by any obstacle) reaches the receiving antenna was considered as the LoS environment and the case in which direct waves do not reach the receiving antenna considered an NLoS environment.

#### 3.1.1. Simulation Model

Simulations were performed using a pre-existing MATLAB 3D model environment that mimics the town of Chicago, shown in Figure 7. 

An overall view of the simulation is shown in Figure 8. The simulation was conducted assuming that the HAPS was 2000 m above the ground and the UAV was 200 m above the ground.

For simplicity, we will use the simulation of the LoS rate over the town to illustrate the simulation. First, random radio sources are generated in the town, as shown in Figure 9. Here, the height of each antenna was set to 10 m.

Next, as shown in Figure 10, a receiving antenna (in this case, a UAV) is generated over the town, the number of direct waves is calculated using a ray tracing simulation, and the LoS rate is calculated by calculating the ratio of the number of direct waves to the number of radio wave sources. In this simulation, the number of radio wave sources is set to 1000. The same simulation is performed for the HAPS.

When the simulation is performed away from the center, receivers are generated in several directions, as shown in Figure 11, and the average value of the LoS ratio obtained from each receiver is used as the LoS ratio.

#### 3.1.2. Results of LoS Rate Measurement Simulation

The results of the simulation conducted above are shown in Figure 12. The LoS ratio of the HAPS was about 100% when the center of the urban area (where the radio source was generated) was directly below the HAPS, and about 60% at a distance of 20 km (elevation angle 45°). On the other hand, when a UAV was used, the LoS ratio was about 30% even when the center of the urban area was directly below the UAV, indicating that the majority of the radio sources were out of sight. Therefore, a specific altitude for the HAPS is considered necessary for position estimation by angle estimation.

### 3.2. Location Estimation Simulation

In order to realize a simulation of radio source location estimation using the HAPS, we used MATLAB to estimate the location of radio sources. Several location estimation methods were discussed in Section 2, but considering the wide search area and required estimation accuracy, the Beamformer method and the MUSIC method were used in our simulation. As a comparison, a case in which location estimation was performed at the altitude of the UAV was also simulated. In our analysis, the location estimation simulation was performed for a non-mobile illegal radio source located outdoors in an urban area, as shown in Figure 7.

The simulation process is shown in Figure 13. (1) A radio source is randomly generated, (2) a ray tracing simulation is performed to obtain information about arriving radio waves (direction of arrival, number of reflections, propagation distance, etc.), and (3) the received signal is generated from the aforementioned information. The direction of arrival is estimated by using each estimation method applied on the received data generated in (3). Then, location estimation is performed using the triangularization method. By comparing the randomly generated coordinates of the radio source with the estimated coordinates, the location estimation of the radio source using the HAPS and other methods is evaluated. Figure 13 shows an overview of the simulation.

#### 3.2.1. Receiving and Transmitting Antenna Configuration

Table 1 shows the antenna specifications for estimating the location of a radio wave source using ray tracing simulation. Hereafter, the transmitting antenna is denoted as Tx and the receiving antenna as Rx.

The receiving antenna was modeled after the cylinder antenna used in the HAPS as a base station for the simulation. The actual prototype cylinder antenna can be found in [15]. The specifications are shown in Table 2.

In this paper, the cylinder antenna consists of N= 36 elements, as depicted in Figure 14, where each element was assumed to be omni-directional and the maximum gain was set to 0 dBi in the simulation.

#### 3.2.2. Radio Propagation Model

This section describes the radio propagation model used in the simulation of radio source location estimation using the HAPS. In this simulation, statistical data were not used in the propagation model because a ray tracing simulation was conducted to determine specific radio waves. The free propagation loss was modeled using Friis formula, as follows [16].
(35)$${PL}_{fsp}=10log_{10}{\left(\frac{4\pi d}{\lambda }\right)}^{2}\left[dB\right]$$
where d and λ are the distance between the transceiver pairs and the wavelength of the considering radio wave, respectively.

The overall path loss can be expressed by Equation (36), where the additional attenuation $L_{ref}$ due to earth and building reflections was calculated using the Fresnel equation. The permittivity and conductivity of the surface materials used in these calculations were taken from the ITU-R study [17]. Assuming that both the ground surface and the building are concrete, the relative permittivity and conductivity were calculated as 5.31 and 0.0548 [S/m], respectively.
(36)$$PL=PL_{fspl}+PL_{ref}\;\;\;\;\;\;\left[dB\right]$$

#### 3.2.3. Signal Processing System

This section describes the signal processing performed in the location estimation simulation. First, the analytical signal input to the cylinder antenna is obtained by generating a single radio source in a 3D model and considering the case where *L* radio waves are incident from the source. The propagation loss $PL_{l}\left[dB\right]\;(l=1,2,\ldots ,L)$ is obtained from Equation (37) using the propagation distance and other data obtained from the ray tracing simulation, and the input power is calculated by following formula.
(37)$$P_{l}=10\times 10^{-\left(\frac{PL_{l}}{10}\right)}\;\;\left[W\right]\;(l=1,2,\ldots ,L)$$

Using this value, the propagation time delay against the direct wave $\tau_{l}(l=1,2,\ldots ,L)$, $(\tau_{1}=0)\;$and the times of reflection $m_{l}(l=1,2,\ldots ,L)$, the complex waveform of the *l*-th wave can be written by the following formula.
(38)$$F_{l}(t)=\sqrt{P_{l}}\exp\left\{j2\pi f\left(t-\tau_{l}\right)\right\}\cdot {\left(-1\right)}^{m_{l}}\;\;\left(l=1,2,\ldots L\right)\;\;$$

The array matrix for the cylinder antennas is then expressed by the following equation [12].
(39)$$A=\left[a\left(\theta_{1},\phi_{1},f\right),\ldots ,a\left(\theta_{L},\phi_{L},f\right)\right]$$
(40)$$a\left(\theta_{l},\phi_{l},f\right)={\left[\exp\left\{j2\pi \frac{f}{c}r_{1}^{T}L\left(\theta_{l},\phi_{l}\right)\right\}\;,\ldots ,\exp\left\{j2\pi \frac{f}{c}r_{N}^{T}L\left(\theta_{l},\phi_{l}\right)\right\}\;\;\;\right]}^{T}(l=1,2,\ldots ,L)$$
(41)$$L\left(\theta ,\phi \right)≜\left[sin\theta cos\phi ,sin\theta sin\phi ,cos\theta \right]$$
where $r_{k}(k=1,2,\ldots ,N)$ is the position vector of each element, and θ and ϕ are the elevation and azimuth angles to the reference point, respectively.

The thermal noise power $P_{N}$ generated at the receiving antenna is given by following equation [18].
(42)$$P_{N}=10\log_{10}\left(kTB\right)+NF\;\;\;\left[dBW\right]\;$$
where k, T, B, NF are Boltzmann’s constant, absolute temperature of the noise source, bandwidth, and noise figure, respectively. The absolute temperature in the HAPS environment was assumed to be 216.5 [K] and the bandwidth was 10 [MHz]. The noise figure was calculated as 3 dB [19]. As a result, the thermal noise power $P_{N}\;$was about −102 [dBm] in the HAPS environment while in the UAV environment, the thermal noise power was slightly higher since the absolute temperature at the UAV altitude is about 288 [K]. Using Equations (38), (39) and (42), the input vector X(t) to each antenna is expressed by the following equation.
(43)$$X\left(t\right)=AF\left(t\right)+N\left(t\right)$$
(44)$$F\left(t\right)={\left[F_{1}\left(t\right),F_{2}\left(t\right),\ldots ,F_{L}\left(t\right)\right]}^{T}$$

In the above equation,$\;N\left(t\right)\;$is the thermal noise vector, whose components are independent complex Gaussian processes with mean zero and variance (power) of $P_{N}$. The input vector, obtained by substituting appropriate values for *t* in Equation (43) for different time samples, was used as the input complex signal for the cylinder array. The number of samples was set to 100 in this paper. By applying the aforementioned Beamformer and MUSIC methods to the obtained input vector X(t), the direction of arrival from the radio source was estimated and used for the location estimation method described in the next section. The direction of arrival with the highest power obtained by the Beamformer and MUSIC methods was used as the estimated direction of arrival.

#### 3.2.4. Location Estimation Method

This section describes a method for obtaining the estimated coordinates of a radio source from the direction of arrival obtained by the MUSIC and Beamformer methods. In this simulation, the direction of arrival of radio waves is estimated from three points as shown in Figure 15. 

The latitude and longitude of each HAPS observation point are converted to the Cartesian coordinate system, and a direction line is created in the x-y coordinate system from the direction of arrival. Figure 16 shows two localization methods from the estimated angular information. In the left-hand side case, all angular information is available from the three sensors, and the center of gravity of the intersections of each directional line is considered as the estimated coordinates of the emitter. In the right-hand side case, the angular information from one of three sensors is missing, so the estimated coordinates are the intersection points of the direction lines of the remaining sensors where angular information is available. For all other cases, the location of the emitter is considered to be unidentified (undetected).

#### 3.2.5. Results of Location Estimation Simulation

The 3D city model shown in Figure 7 was used to randomly generate 1000 radio wave sources, and the location of each source was estimated using the method described in the previous section. The horizontal distance between the center of the urban area with illegal radio sources and the hovering sensors was varied with different scenarios, as seen in our evaluation results below. Three scenarios of sensor altitudes, i.e., HAPS, UAV, and DEURAS, are also separately evaluated in this section.

■HAPS altitude

Table 3 shows the results of localization using the HAPS at three points 5 km away from the urban area. The Beamformer and MUSIC methods were used to estimate the direction of arrival. The estimation probability denotes the percentage that the emitter can be identified (with localization errors) via our proposed system, as mentioned in Figure 16. The stochastics of the estimation errors were derived only from emitters, which were identified by our proposed method. From Table 3, it is observed that there was no significant difference in the results between the Beamformer and MUSIC methods, and 90% of the estimated locations were within 30 m of the true coordinates.

Table 4 shows the results of the estimation using the HAPS at three locations 10 km away from the urban area. Since the sensors are more distanced from the sources compared to the 5 km case, we can observe a slight degradation in terms of localization accuracies as compared to these in Table 3. 

Furthermore, Table 5 shows the results of the estimation using the HAPS at three locations 20 km away from the urban area. Similar to the two aforementioned scenarios, neither the Beamformer nor the MUSIC method showed a significant difference in results. Also, the accuracy of the estimation was poorer at greater distances compared to the cases at 5 km and 10 km. Furthermore, only about 83.6% of the radio transmitting sources in urban areas could be obtained. 

The CDF of localization errors for these scenarios is summarized in Figure 17.

■UAV altitude

Similarly, a simulation was conducted assuming a UAV, with the height of the receiving antenna changed to 200 m. In addition, because the effect of reflections is likely to increase at the altitude of the UAV, the simulation was performed considering radio waves that can be reflected up to four times. Table 6, Table 7 and Table 8 show the results of the estimation using UAVs at three locations (5 km, 10 km and 20 km away from the urban area, respectively). 

In Table 6, only 20% of the total number of radio sources could be identified due to the low LoS ratio, as described in Section 3.1. The localization estimation error of the radio sources that could be identified was also larger than that of the HAPS case. 

Table 7 shows the results of the estimation using UAVs at three locations 10 km away from the urban area. As in the 5 km case, the number of radio sources that could be identified was small. However, the estimation error and the CDF90% value was slightly improved against the 5 km case. This is thought to be due to the fact that the 10 km environment has smaller multipath components compared to the 5 km case, such that angular estimation has higher accuracy. 

Table 8 shows the results of the estimation using UAVs at three locations 20 km away from the urban area. Both the estimated probability and localization accuracy are degraded in this case. It is thought to be due to the fact that the sensors are located too far away, such that the arrival waves after multiple reflections are too weak compared to the noise level. 

The CDF of localization errors for these different scenarios is summarized in Figure 18.

■DEURAS

Finally, a model simulating the DEURAS system was created by installing the receiving antennas on the rooftop of buildings with good visibility in an urban area, as shown in Figure 19. Other system parameters were the same as those for UAVs. 

Table 9 shows the evaluation results of position estimation simulations using the Beamformer and MUSIC methods where DEURAS sensors are located, as shown in Figure 19. Compared to location estimation using UAVs, the estimation accuracy is still poorer, except for the fact that DEURAS can provide a higher estimated probability since the sensors are close to the sources. 

For a fair comparison, we conducted new simulations that the HAPS and UAVs were deployed at three points at a distance of 500 m from the center of the urban area (just right above the area similar to the case of the DEURAS system).

Table 10 shows the evaluation results of the estimation using the HAPS at three locations 500 m away from the urban area. Similarly, Table 11 shows the results of the evaluation when estimating using UAVs at three locations 500 m away from the urban area.

Figure 20 shows a summary of the CDF of localization errors of all scenarios. These results show that the HAPS shows the best performance in terms of both estimated probability and localization accuracy. It is owing to the fact that the HAPS has a higher LoS ratio to guarantee a higher estimated probability, and also direct waves are more likely to arrive at the HAPS rather than multiple reflection paths that increase the angular estimation accuracy.

## 4. Conclusions

In this study, we compared the HAPS, UAV, and DEURAS systems for the location estimation of radio sources, and found that the HAPS estimation showed excellent accuracy at distances up to 20 km due to its high LoS ratio, while the UAV estimation showed poor performance due to interference from reflected waves, especially in urban environments. The position estimation using an antenna configuration that mimics the DEURAS system also showed lower position estimation accuracy compared to the HAPS due to the antenna height and reflected waves. Overall, the HAPS is excellent for monitoring radio sources over a wide area, and we believe that integrating the HAPS into the existing DEURAS setup has the potential to enhance the current system. 

However, several problems are expected to arise when the HAPS is actually used for location estimation in practice. The first is the coordination among three HAPS units. The DEURAS system uses a wide-area cabled LAN to aggregate information from each receiving antenna at the center station, but it is considered difficult to use this method for the HAPS since wired backhauling is impossible for aerial stations. However, such issues can be solved via wireless or optical backhauling among HAPS units, or by using feeder links to coordinate with other HAPSs via ground antennas. The second is the case where there are multiple sources of emitters in the same service area. As a first step to demonstrate the benefit of the proposed system, this paper assumed only a single source. But in realistic circumstances, there could be multiple sources emitted at the same frequency and time due to the nature of illegal emitters. In such a case, the HAPS units need to identify multiple sources simultaneously. This issue can also be solved via sharing the antenna resources equipped on the HAPS units. The third is when the illegal emitter is not fixed is but moving like a car or a truck. In such case, the introduction of a tracking filter upon the mechanism in this paper might be a solution. Although we have not yet been able to study these issues in depth, they will all be considered for our future works.

Furthermore, we used MATLAB to simulate the location estimation of a radio source. It is possible that diffracted and scattered waves, in addition to other arriving and reflected waves, may exist in a real environment, and these may affect the estimation results. Therefore, we would like to conduct actual experiments in the future to confirm the accuracy of the estimation. In addition, we would like to consider the location estimation of low-power radio sources, moving radio sources, and indoor radio sources, which were not covered in this study.

## Figures and Tables

**Figure 1 sensors-24-05803-f001:**
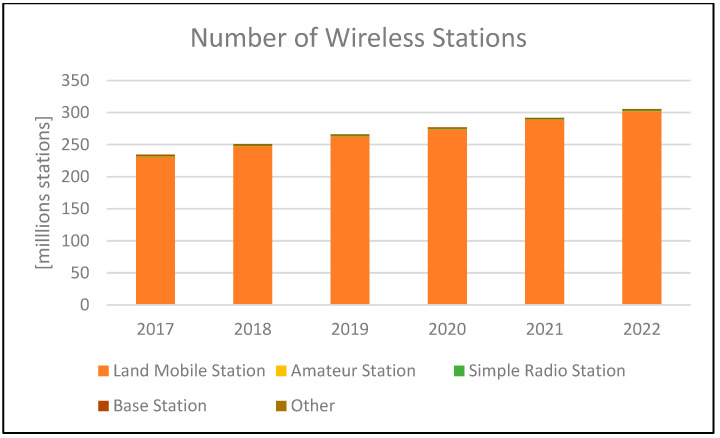
Number of wireless stations in Japan.

**Figure 2 sensors-24-05803-f002:**
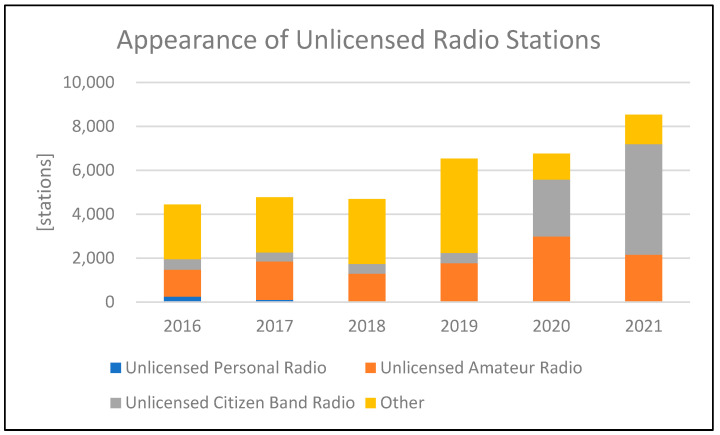
Statistics of illegal radio stations in Japan.

**Figure 3 sensors-24-05803-f003:**
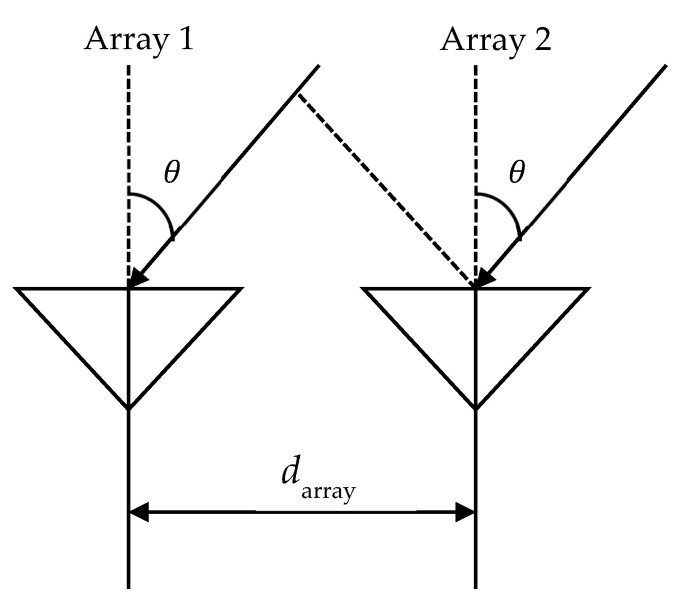
Array antenna.

**Figure 4 sensors-24-05803-f004:**
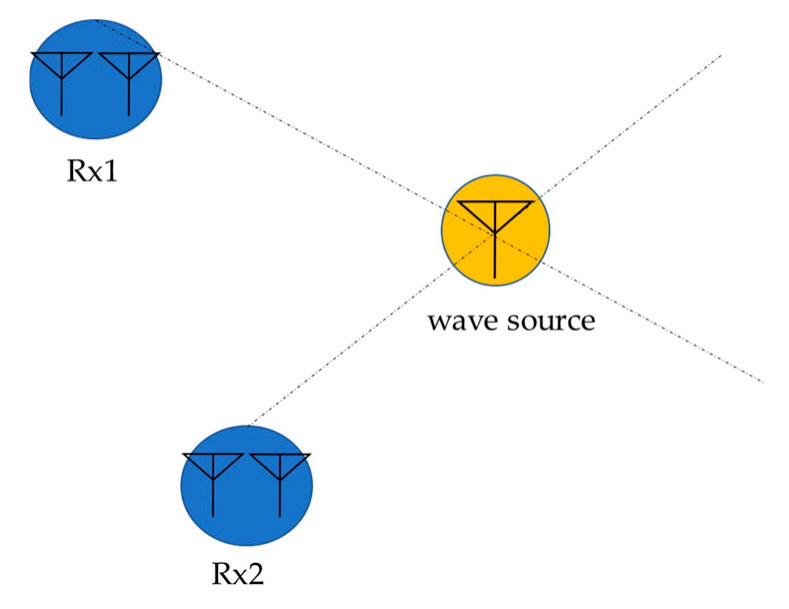
Triangulation method.

**Figure 5 sensors-24-05803-f005:**
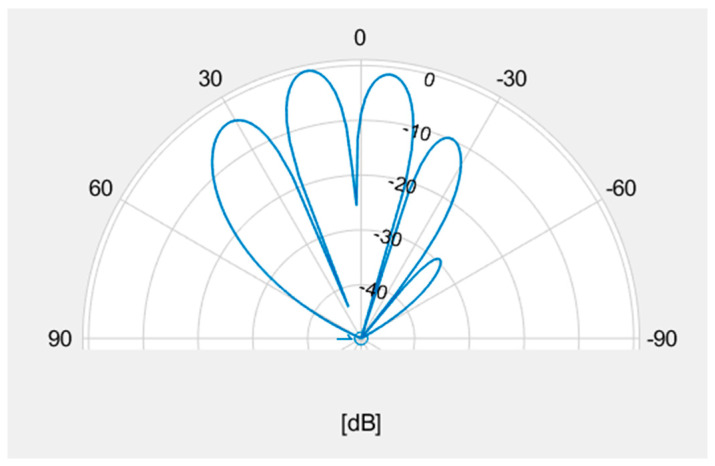
An example of an antenna pattern.

**Figure 6 sensors-24-05803-f006:**
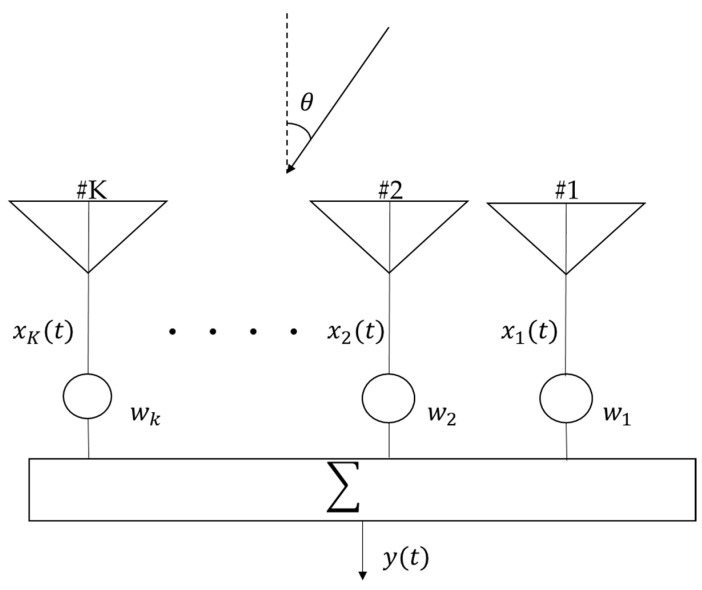
K-element linear array.

**Figure 7 sensors-24-05803-f007:**
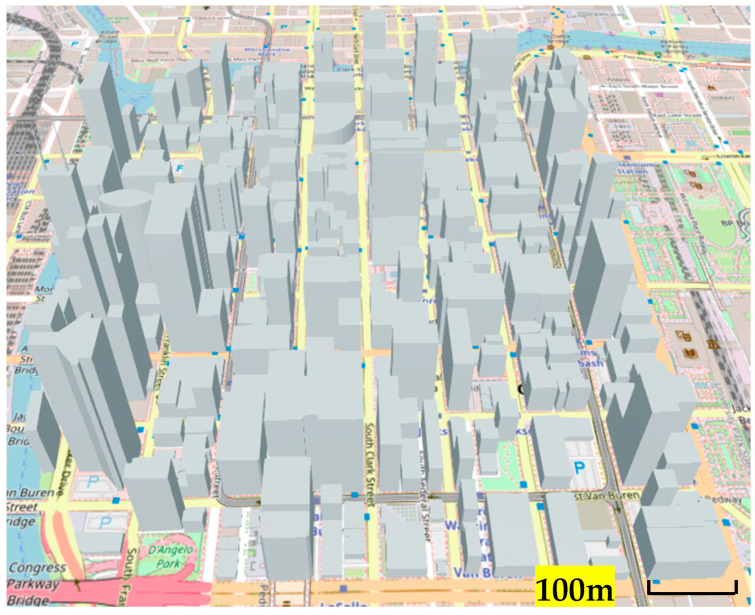
3D model of the evaluation environment.

**Figure 8 sensors-24-05803-f008:**
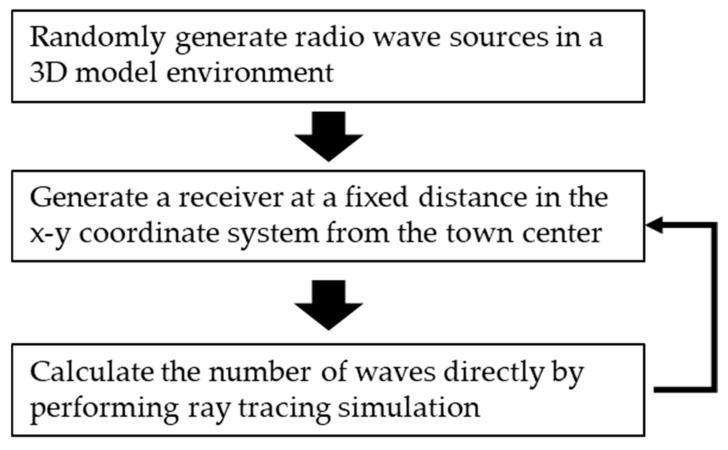
Simulation overview of LoS ratio calculation.

**Figure 9 sensors-24-05803-f009:**
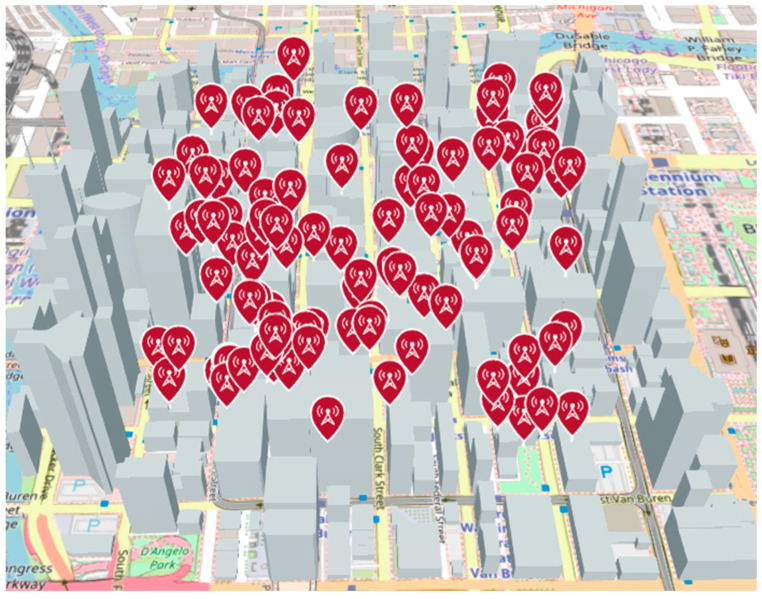
Randomly generated radio wave sources.

**Figure 10 sensors-24-05803-f010:**
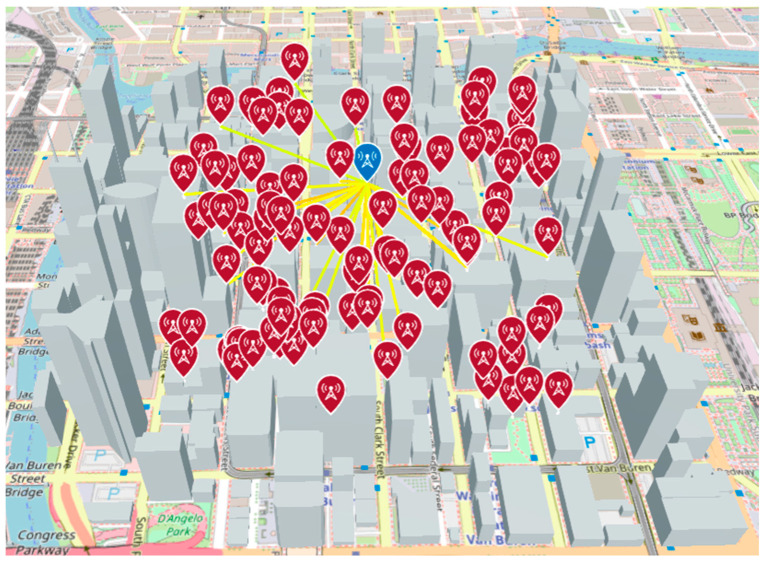
An example of ray tracing simulation in progress.

**Figure 11 sensors-24-05803-f011:**
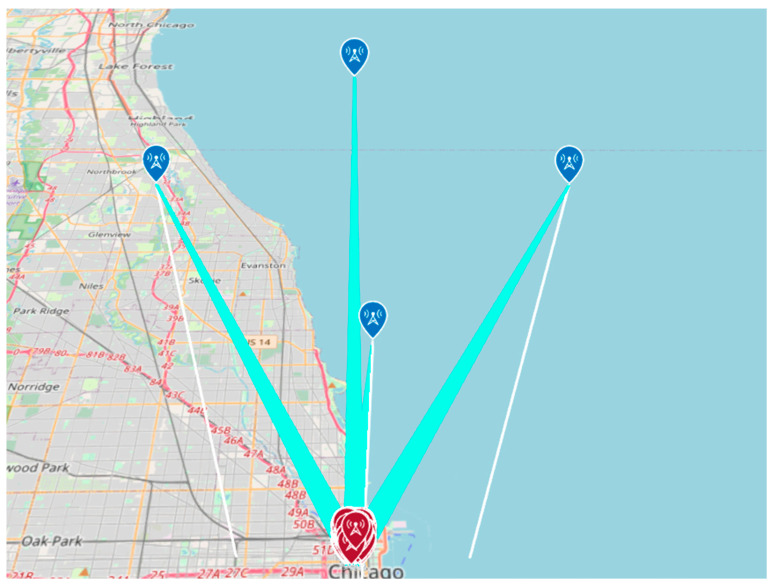
Analysis of LoS ratio when sensors are away from the evaluation center.

**Figure 12 sensors-24-05803-f012:**
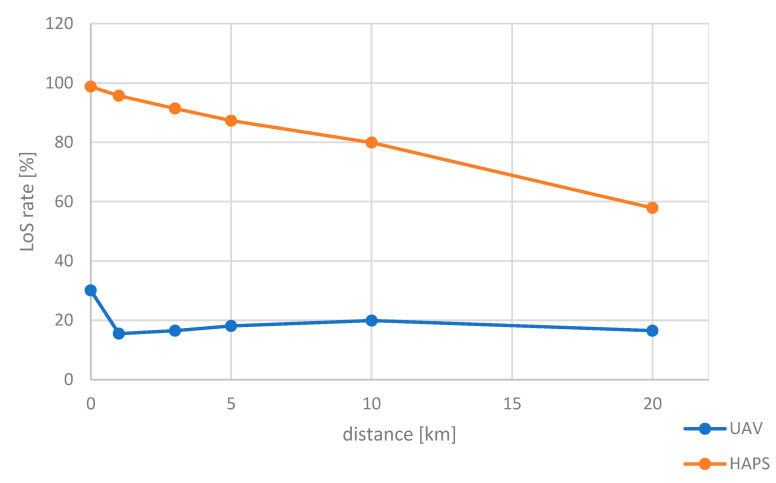
Relationship between distance and LoS ratio.

**Figure 13 sensors-24-05803-f013:**
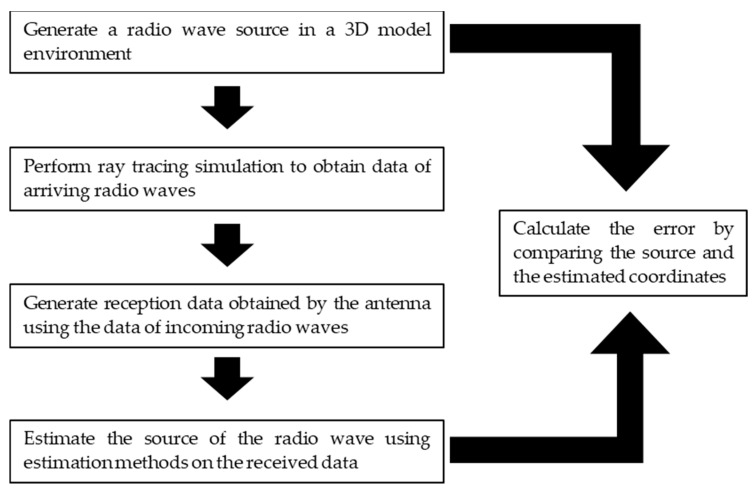
Simulation overview of our location estimation process.

**Figure 14 sensors-24-05803-f014:**
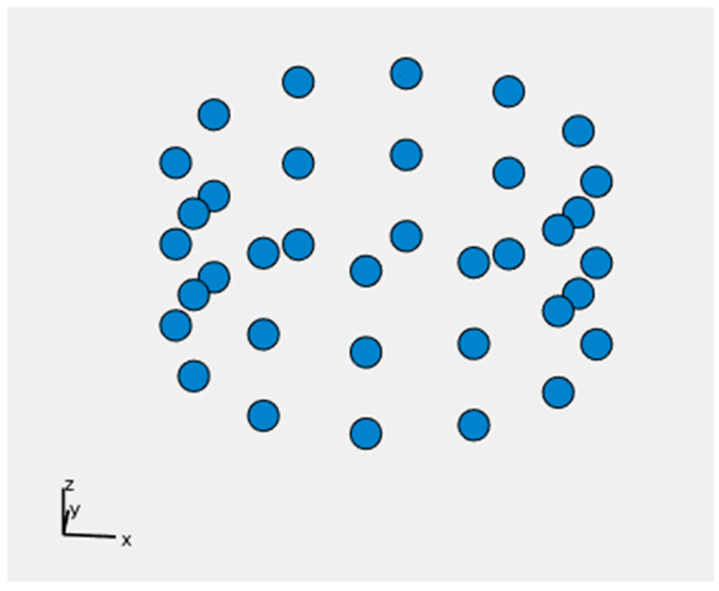
Cylinder antenna used for simulation.

**Figure 15 sensors-24-05803-f015:**
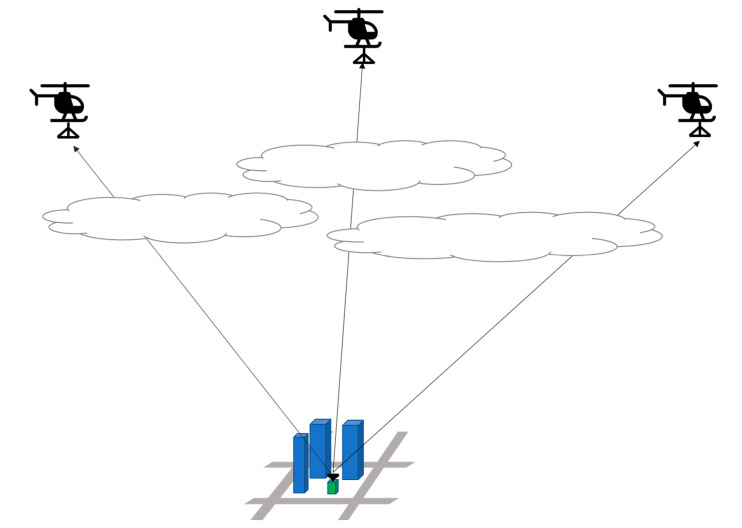
Direction of arrival estimation using high altitude sensors.

**Figure 16 sensors-24-05803-f016:**
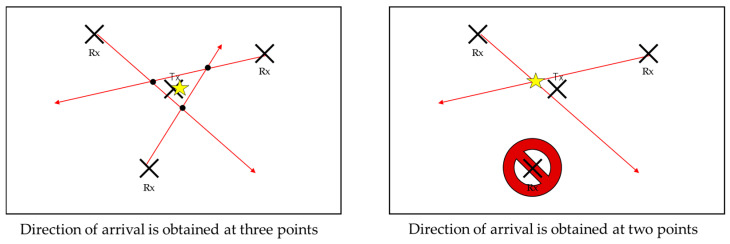
Location estimation method.

**Figure 17 sensors-24-05803-f017:**
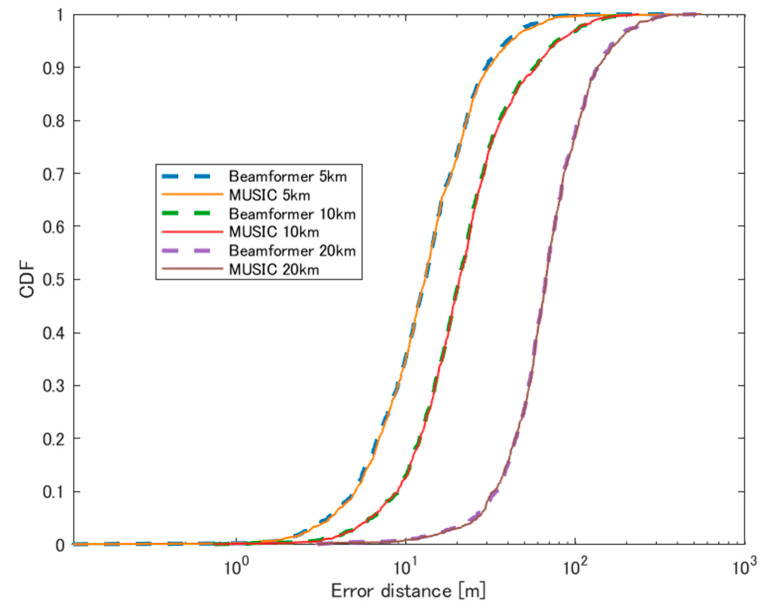
CDF of errors using the HAPS at different distances against the emitting sources.

**Figure 18 sensors-24-05803-f018:**
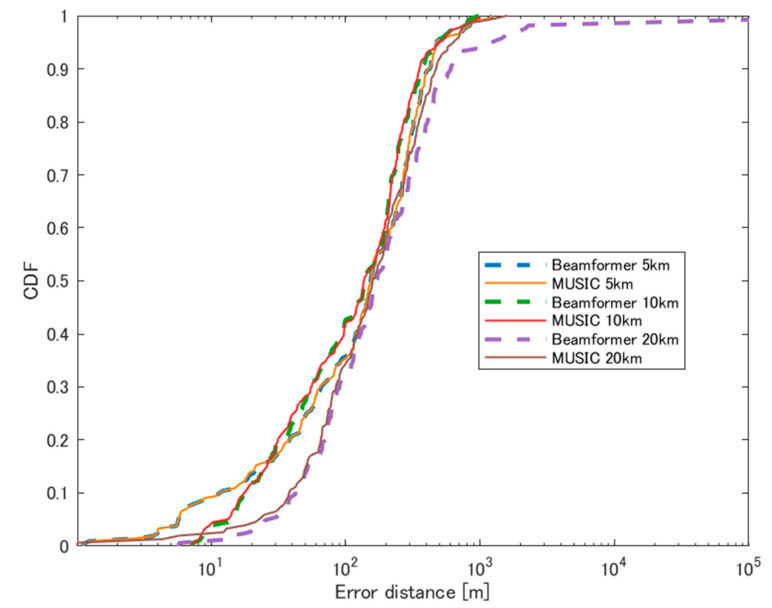
CDF of errors using UAV at each distance.

**Figure 19 sensors-24-05803-f019:**
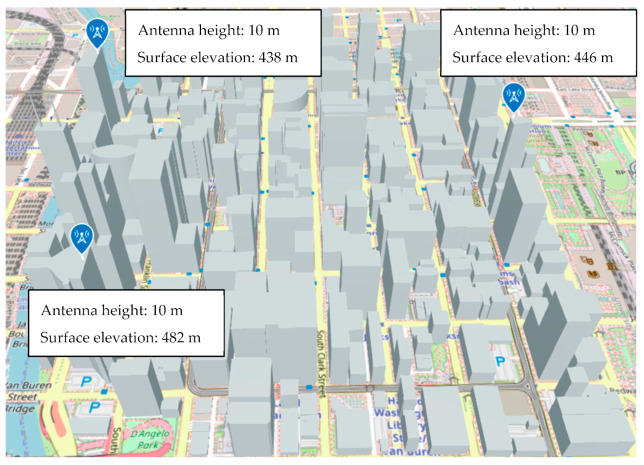
Receiving antenna layout assuming the DEURAS system.

**Figure 20 sensors-24-05803-f020:**
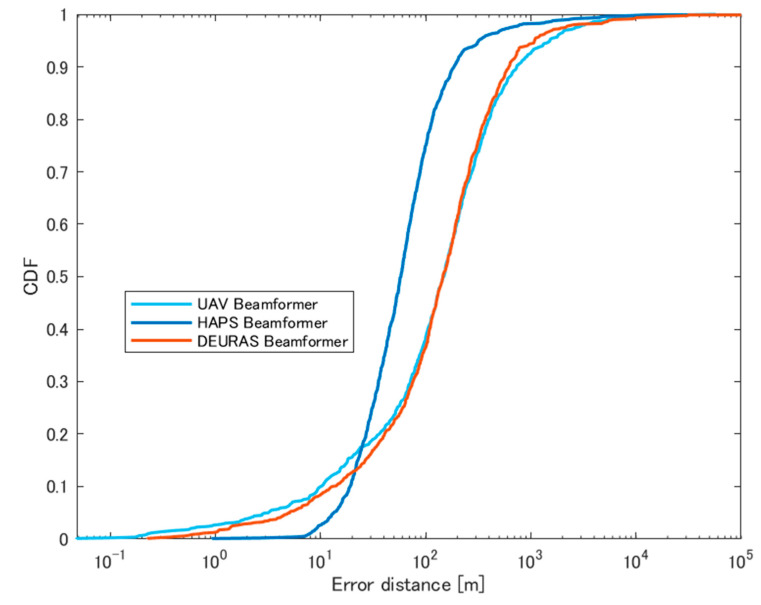
CDF of errors for different sensor altitudes with sensor radius of 500 m.

**Table 1 sensors-24-05803-t001:** Antenna specifications.

Analysis Model		3D (Ray Tracing)
Tx	Center Frequency [GHz]	2
	Frequency Bandwidth [MHz]	10
	Transmission power [W]	10
	Antenna Type	Isotropic
	Antenna Height [m]	10
Rx	Antenna Type	Isotropic
	Antenna Height [m]	200 (UAV)/20,000 (HAPS)
Number of reflections		4 (UAV)/2 (HAPS)

**Table 2 sensors-24-05803-t002:** Specifications of the cylinder antenna.

Direction	Item	Value
Horizontal	Number of elements	12
	Element spacing [m]	0.6λ
Vertical	Number of elements	3
	Element spacing [m]	0.5λ

**Table 3 sensors-24-05803-t003:** Results with the HAPS 5 km away.

	Beamformer	MUSIC
Mean error [m]	16.4	17.0
CDF90% [m]	30.0	30.7
Detection percentage [%]	95.6	

**Table 4 sensors-24-05803-t004:** Results with the HAPS 10 km away.

	Beamformer	MUSIC
Mean error [m]	28.2	28.4
CDF90% [m]	57.8	58.4
Detection percentage [%]	92.1	

**Table 5 sensors-24-05803-t005:** Results with the HAPS 20 km away.

	Beamformer	MUSIC
Mean error [m]	79.0	79.6
CDF90% [m]	133.9	134.5
Detection percentage [%]	83.6	

**Table 6 sensors-24-05803-t006:** Results with UAV 5 km away.

	Beamformer	MUSIC
Mean error [m]	205.1	204.9
CDF90% [m]	424.9	421.5
Detection percentage [%]	21.1	

**Table 7 sensors-24-05803-t007:** Results with UAV 10 km away.

	Beamformer	MUSIC
Mean error [m]	180.2	179.5
CDF90% [m]	365.2	358.9
Detection percentage [%]	17.8	

**Table 8 sensors-24-05803-t008:** Results with UAV 20 km away.

	Beamformer	MUSIC
Mean error [m]	277.8	227.7
CDF90% [m]	607.7	472.6
Detection percentage [%]	16.4	15.2

**Table 9 sensors-24-05803-t009:** Results with antenna placement assuming the DEURAS system.

	Beamformer	MUSIC
Mean error [m]	444.8	444.9
CDF90% [m]	615.1	614.9
Detection percentage [%]	79.9	

**Table 10 sensors-24-05803-t010:** Results with the HAPS 500 m away.

	Beamformer	MUSIC
Mean error [m]	150.7	151.6
CDF90% [m]	186.0	186.1
Detection percentage [%]	98.7	

**Table 11 sensors-24-05803-t011:** Results with UAV 500 m away.

	Beamformer	MUSIC
Mean error [m]	511.0	515.7
CDF90% [m]	735.1	735.1
Detection percentage [%]	79.4	

## Data Availability

Data is contained within the article.

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
