# Peer review of "A Study on a Radio Source Location Estimation System Using High Altitude Platform Stations (HAPS)"

_sensors, 2024, doi:10.3390/s24175803_

Round 1

Reviewer 1 Report

Comments and Suggestions for Authors

In this paper, the author proposes using HAPS for radio wave source estimation and conducts performance evaluations through simulations, comparing it with traditional UAV and DEURAS systems. The simulation results indicate that using HAPS for estimation provides higher estimation probability and accuracy compared to traditional methods. Despite the detailed experimental comparisons presented in this paper, the innovation is somewhat lacking. The paper merely integrates HAPS into the existing DEURAS system. It does not consider potential issues that might arise from using HAPS, such as the possibility of decreased estimation accuracy due to HAPS movement in the air.
Suggestions are as follows:

1. Please cite the relevant literature for all formulas used in the paper. 

2. Please analyze the potential new issues that may arise affecting estimation accuracy when integrating HAPS into the existing DEURAS system, and propose new solutions to address these challenges.

Comments on the Quality of English Language

Minor editing of English language required.

Reviewer 2 Report

Comments and Suggestions for Authors

This paper proposed a radio wave source estimation using HAPS, an unmanned aerial vehicle flying at high altitude. In this study, the authors compared HAPS, UAV and DEURAS systems for location estimation of radio sources, and the simulation results show that estimation from high altitude provides higher estimation probability and estimation accuracy than the conventional method.

The research has certain practical significance, but the paper still has the following problems:

(1) The coordinate axes in Figure 1 lack units. While the horizontal axis is understood to represent years, the absence of units for the vertical axis's numerical values results in inconsistency with the "298.27 million" mentioned in the introduction. The same problem appears in Figure 2.

(2) The introduction focuses on the importance of monitoring illegal radio stations, yet it provides limited information on HAPS and radio wave monitoring technology. It is recommended to add relevant literature.

(3) The text within figures should not be overly large. Specifically, the font size in Figure 3 is too prominent, making the text appear jarring.

(4) Every function and variable introduced should be accompanied by an explanation. The article lacks clarification for Im{} and Re{} in Equation (2). Kindly review other parts of the article for similar omissions.

(5) The variable format in the article should be consistent, but the “K-element” in line 114 and the “K-element” in line 116 are inconsistent. Please scrutinize the article for similar inconsistencies.

(6) It is recommended to add subheadings for the two images on the left and right in Figure 16 to indicate which estimation method the images are from.

(7) The duplication of Figure 18 should be rectified as it constitutes an error in the document's composition.

Comments on the Quality of English Language

No Problem

Round 2

Reviewer 1 Report

Comments and Suggestions for Authors

All changes were made